# AnyEdit++: Adaptive Long-Form Knowledge Editing via Bayesian Surprise

Bowen Tian [* 1]  Caixue He [* 1]  Jiemin Wu [1]  Jingying Wang [2]  Wenshuo Chen [1]  Zexi Li [† 3 4]  Yutao Yue [† 1 5]

## Abstract

Editing complex, long-form knowledge in Large Language Models remains a significant challenge due to the difficulty of maintaining generation coherence. Existing autoregressive methods like AnyEdit alleviate length constraints but rely on Fixed-window Chunking, which disregards logical structure and compromises consistency. To address this, we present **AnyEdit++**, a structure-aware framework incorporating Bayes-Chunk, an adaptive segmentation mechanism that dynamically identifies semantic boundaries based on Bayesian Surprise. We underpin this approach with a theoretical framework establishing two key principles: (1) Structural Independence: we prove that cross-segment interference is minimized when anchor keys are geometrically orthogonal (a condition naturally satisfied by our surprisal-based boundaries but violated by fixed windows), and (2) Causal Locality: we demonstrate that updates injected at these semantic peaks yield strictly superior control compared to arbitrary split points. Extensive experiments across mathematical reasoning, code generation, and narrative tasks demonstrate that AnyEdit++ achieves superior performance and robustness compared to state-of-the-art baselines, validating that structural awareness is critical for effective long-form knowledge editing. Our code is available at Github.

## 1. Introduction

Large Language Models (LLMs) have demonstrated remarkable capabilities in acquiring and storing vast amounts of

---
[*]Equal contribution  [1]The Hong Kong University of Science and Technology (Guangzhou), Guangzhou 511400, China [2]Southeast University, Nanjing, China [3]Knowin AI, Shenzhen, China [4]The Chinese University of Hong Kong, Hong Kong SAR, China [5]Institute of Deep Perception Technology, JITRI, Wuxi 214000, China. Correspondence to: Yutao Yue <yutaoyue@hkust-gz.edu.cn>, Zexi Li <lizexi@knowin.ai>.

*Proceedings of the 43rd International Conference on Machine Learning*, Seoul, South Korea. PMLR 306, 2026. Copyright 2026 by the author(s).

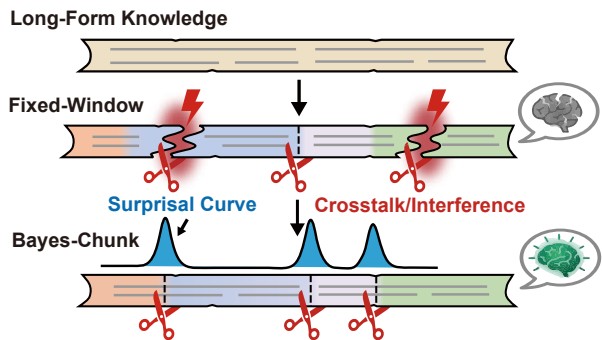

*Figure 1.* Our ***Bayes-Chunk*** method better identifies semantic boundaries to segment long texts for knowledge editing, whereas the ***fixed-window*** segmentation underlying AnyEdit tends to split apart complete semantic paragraphs. This introduces *crosstalk* during the editing process (see subsection 5.1 for details).

knowledge (Achiam et al., 2023; Grattafiori et al., 2024; Tian et al., 2025). However, they inevitably suffer from hallucinations (Zhang et al., 2025) or generate outdated and incorrect information (Meng et al., 2022a). Although fine-tuning can rectify model behavior (Wei et al., 2021), its high computational cost and potential risk of overfitting make **Knowledge Editing** a promising alternative. Traditional *Locate-and-Edit* methods (e.g., ROME (Meng et al., 2022a), MEMIT (Meng et al., 2022b)) typically adopt a paradigm of modifying (usually with added pertubation) the hidden states of a single critical token on a specific layer to maximize the probability of generating the target object, these hidden states are used to guide the updates of the model weights.

While these approaches perform well on simple triplet facts (e.g., the capital of France is Paris, which includes subject, relation, object), they often fail when addressing complex, long-form knowledge involving dense logical dependencies (e.g., mathematical proofs) (Jiang et al., 2025), as single-point updates struggle to support such intensive semantic alterations. This prevents the model from accurately predicting the knowledge to be edited.

To overcome the limitations of single-token editing, recent research introduced the **AnyEdit** (Jiang et al., 2025) framework, which establishes an autoregressive editing paradigm. AnyEdit decomposes long text into continuous segments and achieves knowledge injection through updates under multiple constraints. Despite pioneering long-form editing,

we identify a core design flaw in AnyEdit: it relies on a **fixed-size sliding window** for chunking. This mechanism is oblivious to the distribution of information density within the text, leading to rigid truncations at critical semantic dependencies (e.g., splitting a function definition or separating a mathematical condition from its conclusion, Figure 1). This approach disregards the semantic structure of knowledge, may lead to a decline in editing quality as the injected knowledge gradually becomes more complex.

To address this structural challenge, we propose **AnyEdit++**. Our core insight stems from the observation that the internal state changes of a model when processing sequential data are not uniform; rather, they reflect specific semantic structures through fluctuations in information content. We introduce **Bayesian Surprise** to quantify this process. Intuitively, when an LLM processes a logical turn or the onset of a new semantic unit, its internal representation undergoes a significant shift. Empirically, these high-surprisal points align with natural boundaries in the knowledge structure.

Based on this, AnyEdit++ integrates a plug-and-play adaptive segmentation module named **Bayes-Chunk**. This module dynamically monitors the trajectory of surprisal in the sequence, adaptively aligning segmentation boundaries with positions where the model's hidden states shift significantly. Although the implementation logic of AnyEdit++ is concise, its effectiveness is founded on a *rigorous and systematic theoretical analysis*. Our theoretical framework is primarily constructed from two key perspectives:

**1) Benefits from Structural Independence**: We theoretically justify why our surprise-guided segmentation yields superior editing stability. We prove that cross-segment interference (*crosstalk*) is minimized when the key representations of partitioned chunks are geometrically orthogonal. By aligning with natural semantic boundaries, our Bayes-Chunk strategy ensures this structural independence, whereas fixed-window approaches induce high correlations between segments that theoretically guarantee increased crosstalk.

**2) Causal Locality in Control**: Addressing the question of *where* to edit, we establish the *Principle of Causal Locality*. By analyzing gradient propagation through residual streams versus attention mechanisms, we demonstrate that the immediate predecessor state ($t-1$) offers a control channel strictly superior to distant history. This theoretical insight validates our specific strategy of injecting updates (or perturbations) immediately prior to the **high-surprisal boundaries** identified by Bayes-Chunk, ensuring that maximum control authority is applied precisely when the model's semantic trajectory is at its most critical transition.

To comprehensively validate the performance of AnyEdit++, we conducted **extensive experiments** on multiple mainstream LLMs and benchmarks involving complex logical

tasks (such as mathematical reasoning and code generation). The results indicate that AnyEdit++ not only outperforms all baseline methods (including AnyEdit) on general long-form editing tasks but also achieves significant breakthroughs in editing longer, more logically structured texts.

Our main contributions can be summarized as:

**i) Methodology Contribution**: We propose the AnyEdit++ framework and the Bayes-Chunk algorithm, systematically introducing bayesian surprise into the decision-making process of long-form knowledge editing. This resolves the semantic misalignment caused by fixed windows in a concise and principled manner. **ii) Theoretical Contribution**: We have established a comprehensive theoretical framework that provides a detailed explanation of the principles underlying the effectiveness of our Bayes-Chunk approach. We prove that optimal performance relies on maximizing distinctness between edits (structural independence) and exploiting direct causal links (locality). **iii) Extensive Experiments**: Through extensive experiments across multiple domains and models, we confirm the robustness and superiority of AnyEdit++ in handling complex logical and long-form knowledge, without relying on any complex external auxiliary models.

## 2. Related Work

Knowledge Editing (KE) aims to update specific knowledge in large language models without full retraining, while preserving unrelated behaviors(Sinitsin et al., 2020; De Cao et al., 2021; Li et al., 2026). Existing KE methods are commonly categorized by how the model is modified. Parameter-modifying approaches directly update model weights, including locate-then-edit methods such as ROME (Meng et al., 2022a), MEMIT (Meng et al., 2022b), and WISE (Wang et al., 2024), as well as meta-learning or hypernetwork-based editors like MEND (Mitchell et al., 2022a) and MALMEN (Tan et al., 2024). In contrast, parameter-preserving approaches avoid permanent weight changes by introducing external or auxiliary mechanisms, such as memory-based editing (Mitchell et al., 2022b; Zheng et al., 2023; Zhong et al., 2023) or neuron-augmented methods (Dong et al., 2022; Hartvigsen et al., 2023; Huang et al., 2023). These categories together constitute the dominant methodological landscape of KE.

**From Triples to Arbitrary-Length Knowledge.** Among parameter-modifying methods, Locate-then-Edit (Meng et al., 2022a) has emerged as a core paradigm, initially designed for editing isolated factual triples of the form (subject, relation, object). Early methods such as ROME and MEMIT assume that factual knowledge can be causally localized to specific layers or parameter subspaces, enabling precise single-location edits. However, subsequent work has shown

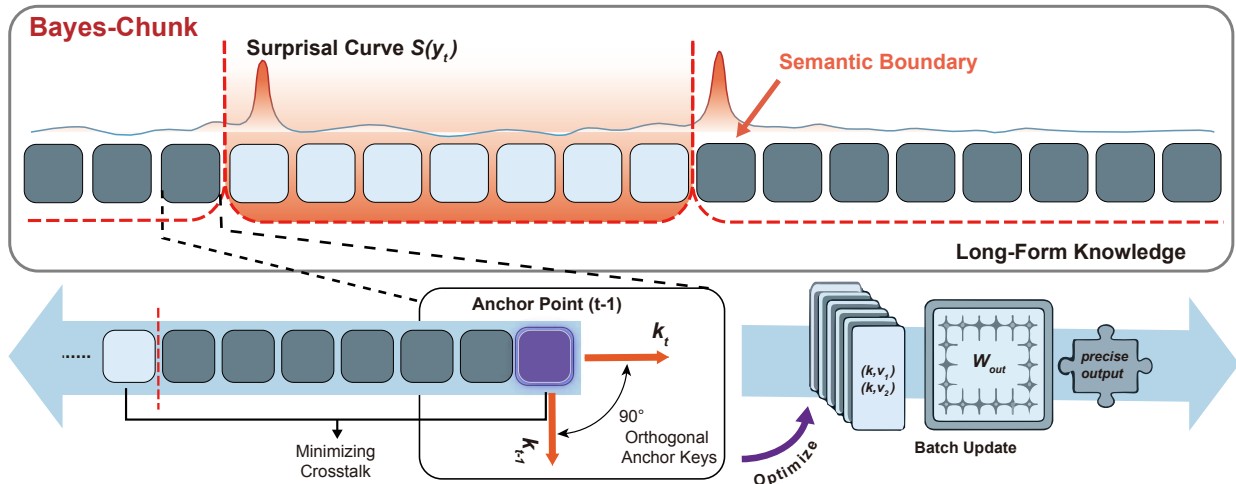

**Figure 2.** This is an overview of our method. When editing long-form knowledge, we compute the Surprisal value for each token in the text, forming the **Surprisal Curve** shown in the figure. We then divide the text into semantic segments based on the Surprisal value's peak points. By injecting perturbations into the hidden state of the token preceding each paragraph, we maximize the generation probability of the current paragraph. Once all perturbations are obtained, they are used to update the model's weights. For details, refer to section 4.

that many forms of knowledge—particularly temporal, commonsense, and conceptual knowledge—are expressed as free-text, multi-token, and distributed representations. Extensions such as METO address temporal knowledge with time-aware objectives (Yin et al., 2023), while MEMITCSK (Gupta et al., 2023) and DEM (Huang et al., 2024) adapt the locate-then-edit framework to commonsense knowledge by tracing multiple tokens or dynamically selecting layers across attention and MLP components. More recently, AnyEdit (Jiang et al., 2025) further relaxes the single-edit assumption by introducing an autoregressive editing paradigm that performs iterative edits over segmented sequences, enabling updates to arbitrarily long and structured texts. However, it relies on a **fixed-size sliding window** for chunking. This mechanism is oblivious to the distribution of information density within the text, leading to rigid truncations at critical semantic dependencies. In contrast, our work introduces a structure-aware adaptive segmentation mechanism that identifies semantic boundaries via Bayesian Surprise, enabling segmentation at high-information transition points and reducing cross-segment interference during editing.

## 3. Preliminaries

In this section, we introduce the prerequisite knowledge required to understand our method.

### 3.1. Standard Knowledge Editing

The objective of knowledge editing is to update an LLM $f_\theta$ to incorporate a specific fact $(s, r, o)$ while maintaining the integrity of unrelated knowledge. Representative *Locate-and-Edit* approaches, such as ROME (Meng et al., 2022a) and MEMIT (Meng et al., 2022b), hypothesize that fac-

tual associations are stored within the dense Feed-Forward Networks (FFNs) of the Transformer. Specifically, these methods target the **output projection** matrix $W_{\text{out}}$ within a critical layer $l$.

The editing process is a two-step procedure comprising *Vector Optimization* and *Weight Modification*.

**Vector Optimization.** First, the method identifies the subject's last token index as the editing location. Within the FFN at layer $l$, the activation vector fed into $W_{\text{out}}$ corresponds to the **key vector** $k \in \mathbb{R}^{d_{\text{inter}}}$. Note that $k$ is derived from the current weights and represents the "trigger" for the factual recall.

To inject the new object $o$, we do not solve for weights directly. Instead, we optimize a **perturbation vector** $\delta \in \mathbb{R}^{d_{\text{model}}}$ that is added to the output of $W_{\text{out}}$. This perturbation shifts the hidden state to maximize the likelihood of the target object $o$:

$$\delta^* = \arg\max_\delta \log P\big(o \mid f_\theta(h_{\text{out}} \leftarrow h_{\text{out}} + \delta); s, r\big),$$

where $h_{\text{out}} = W_{\text{out}}k$ is the original projection output. Consequently, the target value vector for the matrix update becomes $v^* = h_{\text{out}} + \delta^*$. This $v^*$ represents the corrected output direction that the updated $W_{\text{out}}$ should map to when triggered by $k$.

**Weight Modification.** With the target pair $(k, v^*)$ established, the goal is to update the weight matrix $W_{\text{out}}$ to map $k$ to $v^*$ while preserving existing knowledge. This is formu-

lated as a constrained least-squares problem:

$$W_{\text{out}}^* = \arg\min_W \left( \|Wk - v^*\|^2 \right.$$
$$\left. + \lambda \mathbb{E}_{x \sim \mathcal{K}} \|Wk_x - f_{\text{old}}(k_x)\|^2 \right).$$

where $\mathcal{K}$ represents the covariance statistics of keys from general text. The closed-form update is:

$$W_{\text{out}}^* = W_{\text{out}} + \delta^* k^T (C + kk^T)^{-1}.$$

This paradigm is effective for short facts but faces an *efficacy barrier* with long-form content. Encoding a complex narrative into a single perturbation $\delta$ often exceeds the model's capacity, leading to generation collapse (Jiang et al., 2025).

### 3.2. Autoregressive Editing via AnyEdit

To resolve the single-vector bottleneck, AnyEdit (Jiang et al., 2025) extends the paradigm to a sequential autoregressive process. It operates on a *Divide-and-Conquer* principle, decomposing a long sequence $Y$ into shorter segments $\mathcal{C} = \{C_1, \ldots, C_M\}$.

The core framework iteratively applies the standard vector optimization but distributes the load across multiple steps. This involves distinct definitions for **anchoring** (location) and **keys** (representation):

**Sequential Anchoring:** For the $t$-th segment $C_t$, AnyEdit designates the specific token position at the end of the preceding segment $C_{t-1}$ as the **anchor point**. At this token position and target layer $l$, the input vector to $W_{\text{out}}$ is extracted as the local key vector $k_t$.

**Local Perturbation Optimization:** We then optimize a local perturbation $\delta_t$ corresponding to this anchor. The objective is to find a $\delta_t$ which, when added to the FFN output at the anchor point, steers the generation of the specific segment $C_t$. Crucially, this optimization conditions on the history of previous edits:

$$\delta_t = \arg\max_\delta \log P\big(C_t \mid \mathcal{P}, C_{<t}, \Delta_{<t},$$
$$h_{\text{out}}^{(t)} \leftarrow W_{\text{out}} k_t + \delta; \theta\big).$$

Once $\delta_t$ is optimized, we compute the target value $v_t = W_{\text{out}} k_t + \delta_t$. By iterating this process, AnyEdit constructs a dataset of pairs $\mathcal{D}_{\text{edit}} = \{(k_t, v_t)\}_{t=1}^M$ representing the logical chain. Finally, $W_{\text{out}}$ is updated in a single batch using the standard least-squares objective (e.g., MEMIT algorithm) to accommodate all localized edits simultaneously.

**Limitation.** Standard AnyEdit relies on a **Fixed-Window Chunking** strategy (e.g., slicing text every $L$ tokens). This rigid heuristic ignores the semantic topology, often placing anchor points at positions where the key vector $k_t$ is semantically weak or ambiguous. Such suboptimal anchors fail to effectively regulate the subsequent generation, a deficiency our method aims to rectify.

## 4. Method: AnyEdit++

Building upon the previously introduced autoregressive editing framework, we present **AnyEdit++**. While the original AnyEdit validates the efficacy of sequential editing, its performance is hindered by the rigidity of Fixed-Window Chunking. AnyEdit++ retains the core *Divide-and-Conquer* editing pipeline but integrates a novel segmentation mechanism: **Bayes-Chunk**. This module dynamically partitions the target text based on the model's intrinsic belief evolution rather than arbitrary length constraints.

### 4.1. Bayes-Chunk: Surprise-Guided Segmentation

Bayes-Chunk seeks to align segmentation boundaries with the natural event horizons of information flow. We identify these boundaries using **Bayesian Surprise**, which quantifies the shift in the model's understanding upon processing a new token.

#### 4.1.1. MODELING BELIEF STATE DYNAMICS

Let $\pi_t(\cdot) = P(\cdot \mid y_{<t}; \theta)$ denote the model's belief state (probability distribution over the vocabulary) at step $t$. Upon observing the token $y_t$, the belief state updates to $\pi_{t+1}$. The theoretical **Bayesian Surprise** is defined as the Kullback-Leibler (KL) divergence between the prior and posterior belief states:

$$D_{\text{KL}}(\pi_{t+1} \,\|\, \pi_t) = \sum_{w \in \mathcal{V}} P(w|y_{\leq t}) \log \frac{P(w|y_{\leq t})}{P(w|y_{<t})}.$$

In autoregressive language modeling, this divergence is heavily dominated by the probability assigned to the actual observed token. Therefore, we operationalize this metric using the **Information Surprisal** (or Information Content), which serves as a computationally efficient proxy for the instantaneous surprise elicited by token $y_t$:

$$\mathcal{S}(y_t) \approx -\log P(y_t \mid y_{<t}; \theta).$$

A high value of $\mathcal{S}(y_t)$ indicates that $y_t$ conveys high information content, such as a plot twist in a narrative or a logical leap in reasoning, forcing a significant reorganization of the model's hidden representation.

#### 4.1.2. ADAPTIVE BOUNDARY DETECTION

We posit that semantic boundaries correspond to local maxima in this surprise spectrum. To partition the text sequence $Y$ into $M$ cohesive units, we compute the surprise score for all tokens and identify the indices $\mathcal{B} = \{b_1, \ldots, b_M\}$

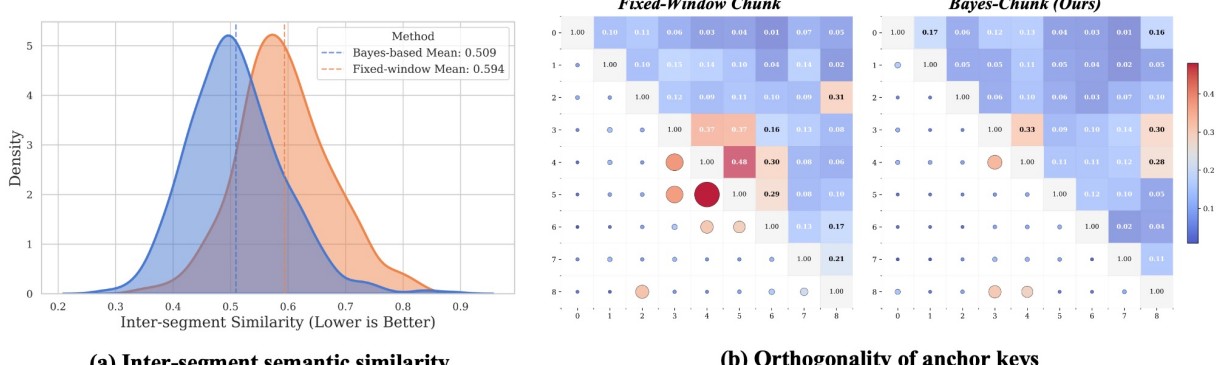

**(a) Inter-segment semantic similarity**

**(b) Orthogonality of anchor keys**

*Figure 3.* To support the conclusion of Theorem 5.1, we demonstrate that our Bayes-Chunk achieves more **independent** segmentation with **minimal** crosstalk by evaluating similarity across two dimensions: semantic and anchor keys extracted from segments.

corresponding to the top-$M$ local peaks:

$$\mathcal{B} = \text{Sort}_{asc}\left(\underset{t\in[1,T]}{\text{argtopk}}\{\mathcal{S}(y_t)\}\right).$$

The text is then segmented into chunks $\mathcal{C} = \{C_1, \ldots, C_M\}$, where the $j$-th chunk spans from the high-surprise token $y_{b_j}$ to $y_{b_{j+1}-1}$. This ensures that each segment encapsulates a complete semantic transition.

### 4.2. Unified Editing with Optimized Anchors

With the semantic boundaries established via Bayes-Chunk, AnyEdit++ seamlessly executes the standard sequential editing protocol described in subsection 3.2. Recall that the AnyEdit framework utilizes the hidden state immediately preceding a segment as the anchor. In our context, since each chunk $C_j$ begins at a high-surprise token $y_{b_j}$, the anchor is naturally assigned to the position $b_j - 1$.

Consequently, the subsequent operations *Vector Optimization* for accumulating perturbations and the final *Weight Modification* remain mathematically consistent with the baseline. The distinction lies solely in the input topology: by strictly following the AnyEdit pipeline on Bayes-Chunked data, we ensure that perturbations are optimized to generate cohesive semantic units rather than arbitrary fragments, while the underlying update mechanism remains unchanged.

## 5. Theoretical Analysis

In order to ground the effectiveness of AnyEdit++ in formal principles, we develop a rigorous theoretical framework addressing two core dimensions of long-form knowledge editing: **update stability** and **control efficiency**. Specifically, our analysis aims to resolve two pivotal challenges: (i) the mitigation of *crosstalk* interference across iterative edit steps to ensure the persistence of prior knowledge segments; and (ii) the identification of optimal injection sites that yield maximum causal influence on the target trajectory.

### 5.1. Benefits from Structural Independence

We analyze the stability of the closed-form solution when aggregating updates from multiple segments. A critical challenge in mass-editing is the *crosstalk* interference, where the update for segment $t$ negatively impacts the preservation of segment $j$.

**Theorem 5.1** (Crosstalk Bound). *Under the least-squares objective for knowledge editing, the reconstruction error (crosstalk) for a specific target segment $j$ induced by other segments is bounded by the sum of pairwise interactions between their anchor keys:*

$$\|\mathcal{E}_{cross}^{(j)}\|_2 \leq \sum_{t\neq j}\|\delta_t\|_2 \cdot \alpha_{tj}, \quad \text{where } \alpha_{tj} \triangleq |k_t^T A k_j|.$$

*Here, $A$ is the precision matrix of the pre-training statistics. Crucially, the interaction term is proportional to the cosine similarity: $\alpha_{tj} \propto CosSim(k_t, k_j)$.*

Detailed proof is provided in Appendix C. The theorem implies that to minimize interference ($\mathcal{E}_{cross} \to 0$), the set of anchor keys $\{k_t\}$ must be as **orthogonal** as possible. If two adjacent chunks define keys that are structurally similar (high cosine similarity), the solver cannot distinguish between them, leading to destructive interference (*overwriting* rather than *updating*) and resulting in poor editing quality.

To empirically validate that Bayes-Chunk fulfills the orthogonality requirement suggested by Theorem 5.1, we evaluate the independence of segments across both semantic and feature spaces on the *EditEverything* dataset. As illustrated in Figure 3 (a), we first measure the global semantic independence by calculating the average inter-segment cosine similarity using `Qwen3-Embedding-0.6B`. The Kernel Density Estimation (KDE) plot reveals that Bayes-Chunk significantly shifts the similarity distribution towards lower values (mean **0.509**) compared to the Fixed-Window approach (mean **0.594**), indicating that our method identifies more semantically distinct boundaries. Further-

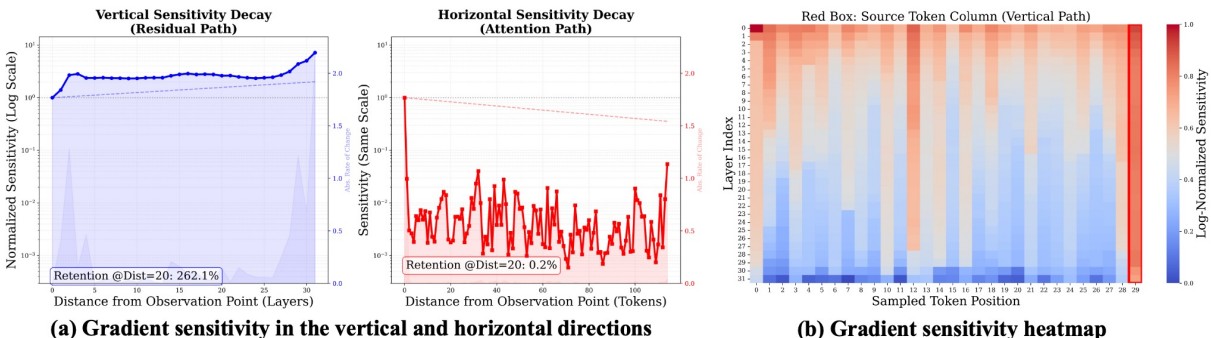

**(a) Gradient sensitivity in the vertical and horizontal directions**   **(b) Gradient sensitivity heatmap**

*Figure 4.* To provide more intuitive evidence supporting the view in Theorem 5.2 regarding gradient sensitivity's **horizontal** and **vertical** propagation within LLMs, we present sensitivity variation patterns and heatmaps for a selected sample across both the token-wise and layer-wise dimensions within the model.

more, we visualize the feature-level correlation of the actual anchor keys $k_t$ extracted from the target layers of Llama-3.1-8B-Instruct. The heatmaps in Figure 3 (b) demonstrate that Bayes-Chunk yields substantially lower pairwise similarity between keys, whereas Fixed-Window exhibits high redundancy between adjacent segments. These results provide direct evidence that our adaptive partitioning minimizes the interaction term $\alpha_{tj}$, thereby effectively suppressing crosstalk to ensure stable and non-destructive knowledge injection (see Appendix D for detailed experimental setups).

## 5.2. Causal Locality in Editing

We first address the fundamental question of *where* to inject the knowledge edits. While sequence models attend to the entire context window, we posit that the direct predecessor state ($h_{t-1}$) offers a strictly superior control channel compared to distant history ($h_{t-k}$).

**Theorem 5.2** (Principle of Causal Locality). *Let $\kappa(i \to t) \triangleq \|\nabla_{h_i}\mathcal{L}(y_t)\|_2$ denote the Positional Controllability, representing the sensitivity of the target prediction $y_t$ to the hidden state at position $i$. For a high-surprisal target, assuming a standard residual network architecture, the controllability decreases significantly as the distance $k$ increases. Specifically, the relative control gain is strictly positive:*

$$\Delta\kappa_k \triangleq \kappa(t-1 \to t) - \kappa(t-k \to t) > 0, \quad \forall k > 1.$$

The proof (detailed in Appendix E) relies on decomposing the gradient flow into **vertical** and **horizontal** components. The vertical propagation through the residual stream at $t-1$ acts as a *quasi-isometry*, preserving the magnitude of the error signal. In contrast, the horizontal propagation to $t-k$ must traverse the attention mechanism, which acts as a bottleneck by distributing the signal weight across the context window (where typically $A_{t-1,t-k} \ll 1$).

To empirically verify the *vertical vs. horizontal* gradient flow dynamics described in Theorem 5.2, we conduct a sen-

sitivity analysis on Llama-3.1-8B-Instruct by back propagating the loss from the last token's hidden state. As illustrated in Figure 4 (a), the gradient sensitivity across model layers (left subplot) remains remarkably stable, exhibiting a quasi-isometry property that preserves signal magnitude throughout the residual stream. In contrast, the sensitivity exhibits a sharp exponential decay as the relative distance from the target token increases (right subplot). This trend is further consolidated by the layer-token sensitivity heatmap (Figure 4 (b)), where high controllability is strictly localized within the most recent token positions across all layers. These observations confirm that $h_{t-1}$ serves as the optimal control channel, as distant tokens suffer from severe horizontal signal attenuation through the attention layers. (Detailed methodology is provided in Appendix F).

## 6. Experimental Results

### 6.1. Results of the comparative experiment

In this section, we primarily present the comparison results between AnyEdit++ and baseline methods on a wide range of knowledge editing datasets.

**Baseline Methods**: The baseline methods we used for comparison include traditional knowledge editing approaches designed for triples, as well as methods specifically engineered for diversifying knowledge in long-form knowledge editing. For the traditional knowledge editing methods, we selected **MEMIT** (Meng et al., 2022b) and **AlphaEdit** (Fang et al., 2024). MEMIT is a classic and widely applicable method in the field of knowledge editing, while AlphaEdit represents a recent key state-of-the-art approach. For long-form text knowledge diversification, we chose **AnyEdit** (Jiang et al., 2025) as the primary comparison target.

**Datasets**: To comprehensively evaluate AnyEdit++ across a spectrum of semantic complexity, ranging from atomic fact updates to long-form structural reasoning, we utilize three distinct benchmarks. We employ **CounterFact** (Meng et al., 2022a) and **UnKE** (Deng et al., 2024) as standard baselines to assess efficacy on atomic facts and unstruc-

*Table 1.* We evaluated the performance of various models and different approaches on knowledge editing using **EditEverything**. We use **bold** text to indicate optimal results, underline to denote suboptimal results, and highlight improvements exceeding 5% in red

| LLM | Method | EditEverything | | | | | | | | | | | | | | Average Total | |
| | | Math | | Code | | Physics | | Chemistry | | Biology | | News | | Poetry | | | |
| | | BLEU | BS | BLEU | BS | BLEU | BS | BLEU | BS | BLEU | BS | BLEU | BS | BLEU | BS | BLEU | BS |
| LLAMA-3.1 8B-Instruct | MEMIT | 41.06 | 89.06 | 35.06 | 77.26 | 57.48 | 88.94 | 56.74 | 91.59 | 55.63 | 87.87 | 25.11 | 74.74 | 27.21 | 69.70 | 42.61 | 82.74 |
| | AlphaEdit | 38.89 | 88.28 | 27.09 | 75.03 | 54.00 | 87.85 | 52.05 | 90.66 | 57.27 | 88.58 | 26.13 | 74.46 | 27.66 | 68.52 | 40.44 | 81.91 |
| | AnyEdit | 74.19 | 95.84 | 81.20 | 95.75 | 79.25 | 96.73 | 81.35 | 97.46 | 78.64 | 96.37 | 64.81 | 91.69 | 49.07 | 85.80 | 72.64 | 94.23 |
| | **Ours** | **77.74** | **96.49** | **86.48** | **97.27** | **80.24** | **97.25** | **81.49** | 97.16 | **80.35** | **96.86** | **68.45** | **91.70** | **50.21** | 84.78 | **75.00** | **94.50** |
| LLAMA-2 7B | MEMIT | 26.36 | 72.90 | 20.77 | 72.97 | 33.20 | 77.33 | 30.39 | 81.21 | 39.28 | 74.11 | 28.50 | 62.71 | 8.01 | 35.86 | 26.64 | 68.16 |
| | AlphaEdit | 29.70 | 73.91 | 10.95 | 62.26 | 35.97 | 80.11 | 29.92 | 78.67 | 39.11 | 77.06 | 26.74 | 74.38 | 9.27 | 48.57 | 25.95 | 70.71 |
| | AnyEdit | 43.05 | 88.23 | 50.78 | 88.62 | 49.20 | 90.37 | 47.00 | 91.77 | 51.82 | 90.32 | 34.34 | 84.03 | 19.94 | 70.99 | 42.30 | 86.33 |
| | **Ours** | **47.31** | **88.52** | **70.50** | **92.80** | **53.83** | **90.44** | **52.00** | **92.65** | **55.39** | **90.51** | **44.85** | **84.91** | **27.04** | **73.77** | **50.13** | **87.66** |
| QWEN-2.5 7B-Instruct | MEMIT | 75.91 | 95.18 | 51.84 | 81.49 | 64.28 | 91.37 | 67.10 | 94.44 | 64.79 | 91.37 | 45.13 | 79.01 | 40.22 | 71.96 | 58.47 | 86.40 |
| | AlphaEdit | 81.18 | 95.76 | 60.74 | 83.42 | 66.87 | 91.51 | 72.57 | 95.46 | 65.70 | 91.79 | 50.17 | 82.09 | 39.78 | 71.17 | 62.43 | 87.31 |
| | AnyEdit | 89.07 | 98.09 | 83.34 | 95.69 | 87.29 | 98.04 | 90.46 | 98.54 | 85.82 | 97.04 | 80.09 | 94.79 | 56.63 | 84.77 | 81.81 | 95.28 |
| | **Ours** | **92.80** | **98.79** | **91.56** | **98.41** | **87.91** | **98.07** | **91.61** | 98.10 | 85.29 | 96.67 | 79.97 | **95.30** | **68.14** | **88.67** | **85.33** | **96.29** |

tured question-answering. To rigorously test coherency in complex long-context generation, we utilize **EditEverything** (Jiang et al., 2025), which encompasses diverse tasks across seven domains (e.g., Math, Code). This combination verifies that our approach maintains high precision on fundamental editing tasks while significantly outperforming baselines in scenarios requiring logical retention and structural consistency. Detailed statistics and descriptions are provided in Appendix G.

*Table 2.* Performance comparison on Reference Benchmarks.

| Method | UnKE | | CounterFact | | Average | |
| | BLEU | BS | BLEU | BS | BLEU | BS |
| MEMIT | 24.76 | 76.50 | 32.21 | 75.79 | 28.49 | 76.15 |
| AlphaEdit | 21.34 | 73.86 | 23.51 | 72.42 | 22.43 | 73.14 |
| AnyEdit | 79.02 | 95.88 | 86.27 | 97.85 | 82.65 | 96.87 |
| **Ours** | **81.57** | **96.03** | **90.69** | **98.29** | **86.13** | **97.16** |

**Evaluation Metric**: To evaluate the strengths and weaknesses of different methods from a more comprehensive perspective, we selected the BLEU and BERT Score (based on all-MiniLM-L6-v2) as our primary observation metrics. Specifically, the BLEU calculation process favors exact matching, enabling assessment of the accuracy of edited model responses relative to standard answers. In contrast, BERT Score focuses more on the semantic similarity between model responses and standard answers. Selecting these two metrics ensures a comprehensive understanding of the edited model responses' strengths and weaknesses, covering both key information matching and semantic levels.

As shown in Table 1, we conducted comprehensive comparative experiments on EditEverything and published results across all subcategories to facilitate observation of trends across different classifications (Since both our approach and AnyEdit are plug-and-play methods, for fairness, we both specified the *MEMIT* algorithm as the base algorithm for integration. For more detailed experimental settings, please refer to Appendix H). It is evident that we achieved the highest average metrics for both BLEU and BERT Score across the three LLM models. Compared to AnyEdit, our approach demonstrated significant gains in BLEU scores: a 2.36% improvement on Llama-3.1-8B-Instruct, a 3.52% increase on Qwen-2.5-7B-Instruct, and nearly an 8% boost on Llama-2-7B. Regarding BERT Score, AnyEdit already achieved solid results using a fixed-window strategy (e.g., 94.5%, 96.29%). Nevertheless, our method continues to show an upward trend, achieving improvements of 0.27%, 1.33%, and 1.01% across the three LLMs, respectively. This demonstrates that our approach consistently outperforms AnyEdit across diverse editing data types and generally longer editing lengths, while requiring only the additional computation of Bayesian Surprise values for edited data.

Specifically, we observed an additional phenomenon: our approach achieved significant improvements under the Math and Code categories. For instance, in the Code classification task, our method outperformed the AnyEdit approach by nearly 20% on Llama-2-7B, while also achieving 5% and 8% gains on the other two models, respectively. This observation suggests that segmentation based on Bayes-Chunk appears to yield greater advantages on data with strong logical structure and longer lengths. We will discuss this phenomenon in detail in subsection 6.2.

To ensure a comprehensive evaluation, we conducted assessments not only on the EditEverything dataset, which is specifically designed for long-form knowledge editing with diverse content, but also on the more traditional UnKE and CounterFact datasets. As shown in Table 2, using Llama-3.1-8B-Instruct as the foundational LLM for editing, our method clearly achieves the best performance on both metrics across different datasets. Compared to the AnyEdit method, it improves BLEU by nearly 4% on average across both datasets while consistently enhancing the BERT Score metric. This result further demonstrates the adaptability of our Bayes-Chunk approach across diverse datasets.

## 6.2. Experiments on data with higher logical complexity and ultra-long lengths

Based on the extensive comparative experiments conducted earlier, we observed that the Bayes-Chunk segmentation approach demonstrates greater advantages when applied to

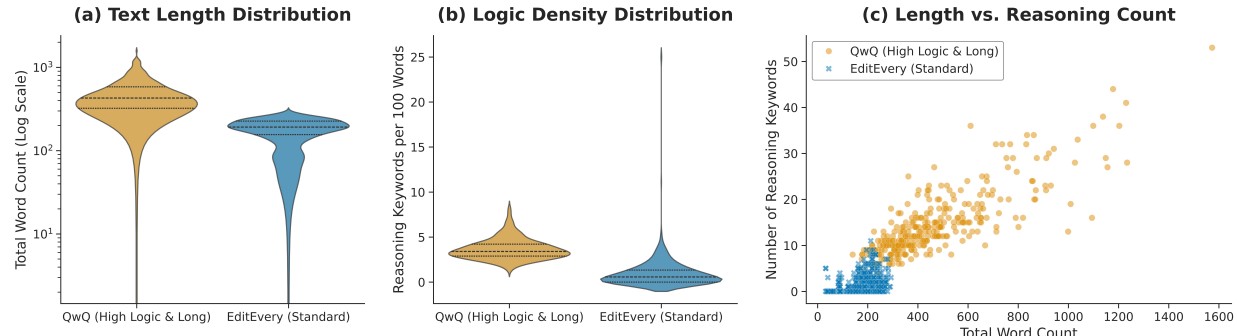

*Figure 5.* Distribution Differences Between the **EditEverything** and Our **QwQ-Edit** in Sample Length and Logical Density

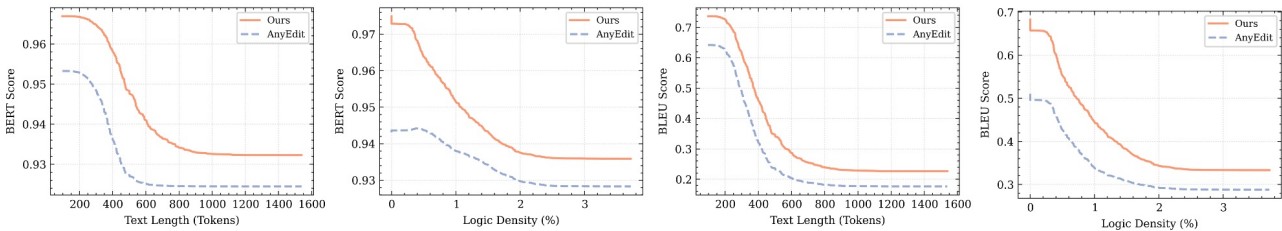

*Figure 6.* We evaluated the AnyEdit method and our approach on QwQ-Edit, reporting the trends in BLEU and BERT Score metrics through fine-grained grouping based on text length and logical density within the dataset.

data with stronger logical coherence and longer segments (multiple-segment data). Incorporating highly logical and lengthy textual data into the model represents a significant developmental direction for the field of knowledge editing. To further investigate this phenomenon, we introduced a new dataset, ***QwQ-Long-CoT-Math-GSM-v1*** (Team, 2025), as our editing dataset. This dataset contains a large number of mathematical problems along with corresponding answers incorporating CoT, featuring relatively high average answer lengths. Due to the substantial size of the complete dataset, we selected a subset to form our actual working dataset (refer to Appendix I for the specific processing workflow). We denote this dataset as **QwQ-Edit**.

To visually illustrate the differences between our constructed **QwQ-Edit** dataset and the **EditEverything** dataset, we present comprehensive statistical graphs of the dataset as shown in Figure 5. Specifically, we compare the two datasets from two perspectives: text length and logical density. Regarding text length comparison, as shown in the density plot in Figure 5 (a), most samples in our QwQ-Edit dataset (high-density regions) exceed the maximum sample length in EditEverything. For logical density, we defined a set of semantically relevant terms (see Appendix I for details) and calculated their density across each dataset's samples to estimate logical coherence. Figure 5 (b) demonstrates that QwQ-Edit consistently maintains higher logical density than EditEverything, with Figure 5 (c) providing a more comprehensive distribution view. Based on these observations, we can demonstrate to a certain extent that QwQ-Edit is a knowledge editing dataset encompassing

more complex logic and longer text lengths.

To verify whether our method achieves a consistent improvement over AnyEdit on the QwQ-Edit dataset, we evaluated both AnyEdit and our method uniformly on this dataset (given the great difficulty of the task, we selected `Qwen-2.5-7B-Instruct` as our target LLM due to its superior performance on logical reasoning tasks). The results are shown in Figure 6. We categorized the samples in the dataset based on length and logical density (details in Appendix I) and plotted the trends of BERT Score and BLEU metrics for both our method and AnyEdit. It is evident that regardless of changes in the horizontal axis, our method **consistently outperforms** AnyEdit on **both** metrics. This fully corroborates our observations from comparative experiments: when editing longer or more logically coherent texts, the Bayes-Chunk-based segmentation approach yields superior editing performance.

### 6.3. Analysis of Fine-Tuning-Based Editing Methods

Recent studies have also explored knowledge editing through fine-tuning (Xiong et al., 2025). However, when handling long-form editing targets, these methods usually still rely on fixed-length segmentation, making it difficult to adaptively determine editing boundaries according to the content. To verify the generality of our segmentation strategy, we combine Bayes segmentation with the fine-tuning-based FT-UKE method and report the results in Table 3. As shown in the table, on both `Llama-3.1-8B-Instruct` and `Qwen-2.5-7B-Instruct`, even when the original FT-UKE already achieves strong performance, introducing

*Table 3.* Editing results of FT-UKE and FT-UKE with Bayes segmentation. The light-blue background indicates results with Bayes segmentation, and bold values indicate improvements over the original FT-UKE based on unrounded scores.

| LLM | Method | EditEverything | | | | | | | | | | | | | | Average | |
|---|---|---|---|---|---|---|---|---|---|---|---|---|---|---|---|---|---|
| | | Math | | Code | | Physics | | Chemistry | | Biology | | News | | Poetry | | Total | |
| | | BLEU | BS | BLEU | BS | BLEU | BS | BLEU | BS | BLEU | BS | BLEU | BS | BLEU | BS | BLEU | BS |
| LLAMA-3.1 8B-Instruct | FT-UKE | 99.89 | 99.97 | 99.87 | 100.00 | 99.83 | 99.99 | 99.98 | 100.00 | 99.98 | 99.99 | 100.00 | 100.00 | 100.00 | 100.00 | 99.90 | 99.99 |
| | FT-UKE + Bayes | **99.97** | **99.98** | **99.88** | 100.00 | **99.91** | 99.99 | **99.99** | 100.00 | **99.99** | **100.00** | 100.00 | 100.00 | 100.00 | 100.00 | **99.95** | **99.99** |
| QWEN-2.5 7B-Instruct | FT-UKE | 98.84 | 99.74 | 99.97 | 100.00 | 99.46 | 99.96 | 99.78 | 99.97 | 99.73 | 99.94 | 99.36 | 99.95 | 100.00 | 100.00 | 99.52 | 99.93 |
| | FT-UKE + Bayes | **99.76** | **99.96** | 99.79 | 99.97 | **99.51** | **99.97** | 99.05 | 99.91 | **99.93** | **99.98** | **99.99** | **100.00** | 98.60 | 99.90 | **99.57** | **99.96** |

---

**Case 1: Syntactic Dependencies (Sample ID 1)**

**Fixed-Window:** ...since 6 is the  product of  — **2** and 3. The largest...

*(Analysis: Splitting inside a prepositional phrase "product of ... 2" creates high semantic dependency.)*

**Bayes-Chunk:** ...since 6 is the  product of 2 and 3.  — **The** largest three-digit...

*(Analysis: Boundary aligns with the end of the logical clause.)*

---

**Case 2: Symbolic Integrity (Sample ID 5)**

**Fixed-Window:** ...Now we can solve for n: ... n = 34  /  — **2**

*(Analysis: The operand "2" is separated from the operator, causing potential hallucination in calculation.)*

**Bayes-Chunk:** ...Now we can solve for n: ... n = 34 / 2  — **n** = 17

*(Analysis: The full arithmetic expression is maintained intact.)*

*Figure 7.* **Comparison of Segmentation Boundaries.** We highlight critical failures in Fixed-Window chunking (red text indicates broken context) versus the adaptive boundaries identified by AnyEdit++ (green text indicates coherent continuations). The vertical bar (—) denotes the exact segmentation point.

Bayes segmentation further improves the overall BLEU and BS scores. This demonstrates that our proposed segmentation mechanism can serve as a plug-and-play module to enhance fine-tuning-based knowledge editing methods.

## 7. Case Study

To visually illustrate the difference between our Bayes-Chunk and AnyEdit's fixed-window approaches, we extracted examples of both segmentation methods from EditEverything as shown in Figure 7. The Sample ID directly corresponds to the sample's ID in the EditEverything dataset, and the segmentation hyperparameter settings align with those used in actual experiments. As demonstrated by the two cases presented, our Bayes-Chunk approach produces more interpretable and logical paragraph divisions when processing identical sample content. This superior paragraph segmentation enables our method to better support editing of lengthy texts and complex logical structures.

## 8. Conclusion

In this paper, we focus on long-form, diverse knowledge editing tasks. Building upon AnyEdit's sliding window approach, we partition longer texts into multiple segments. For each segment, we compute the expected Hidden State at the target position. The weight changes for the target are then derived by jointly constraining the Hidden States computed across multiple segments, enabling knowledge encoding into the LLM's weight space. Unlike AnyEdit, we adopt a Bayesian Surprise–based segmentation strategy. By leveraging the LLM's inherent modeling capabilities to compute the Bayesian Surprise distribution of the data to be edited, we identify higher Bayesian Surprise values as segmentation boundaries. This yields more independent and reasonable paragraph divisions, and we theoretically demonstrate the rationale behind Bayes-Chunk for enhancing editing efficiency, validated through extensive experimentation.

## Acknowledgements

This work was supported by the Guangdong Basic and Applied Basic Research Foundation (Grant No. 2026A1515011579), the HKUST-HKUST(GZ) 1+1+1 Joint Funding Program (Grant No. C_2025_031), and the Guangzhou-HKUST(GZ) Joint Funding Program (Grant No. 2023A03J0008), Education Bureau of Guangzhou Municipality. This work was also supported by Jiangsu Industrial Technology Research Institute (JITRI) and Wuxi National High-Tech District (WND).

## Impact Statement

This paper presents work whose goal is to advance the field of Machine Learning. There are many potential societal consequences of our work, none of which we feel must be specifically highlighted here.

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

# A. Notation

*Table 4.* Summary of mathematical notations used in this paper.

| Symbol | Description |
|---|---|
| $k_t$ | Anchor key representation at anchor position $t$ |
| $v_t$ | Target value corresponding to the edited key $k_t$ |
| $\delta_t$ | Optimized perturbation applied to $k_t$ during editing |
| $W_{\text{out}}$ | Output projection matrix of the language model |
| $D_{\text{edit}}$ | Edit dataset consisting of key–value pairs $\{(k_t, v_t)\}_{t=1}^{M}$ |
| $M$ | Number of anchor points in a document |
| $L$ | Fixed window length in fixed-window chunking |

# B. Terminology

*Table 5.* Clarification of key terms used in this paper.

| Term | Explanation |
|---|---|
| Anchor Key | A key representation selected as the anchor point for a localized model edit. |
| Cross-segment interference (Crosstalk) | Unintended interaction where edits applied to one text segment affect other segments. |
| Surprisal-based boundary | A segmentation boundary determined by peaks in token-level surprisal. |
| Structural Independence | Geometric orthogonality between anchor keys that minimizes edit interference. |
| Causal Locality | The principle that edits injected at semantically salient positions yield superior control. |

## C. Proof of Structural Independence (Theorem 5.1)

In this section, we derive the closed-form solution for the multi-segment editing objective and explicitly bound the crosstalk error.

### C.1. Optimization Objective

We formulate the editing of $M$ segments as a constrained Ridge Regression problem. We seek an update $\Delta W$ that maps validation keys $K = [k_1, \ldots, k_M]$ to target residual vectors $D = [\delta_1, \ldots, \delta_M]$, while preserving general knowledge defined by covariance $C$:

$$\min_{\Delta W} \mathcal{J}(\Delta W) = \|\Delta W K - D\|_F^2 + \lambda \text{Tr}(\Delta W C \Delta W^T).$$

### C.2. Derivation of the Update Rule

Since $\mathcal{J}$ is strictly convex, we find the global minimum by setting the gradient to zero:

$$\frac{\partial \mathcal{J}}{\partial \Delta W} = 2(\Delta W K - D)K^T + 2\lambda \Delta W C = 0.$$

Solving for $\Delta W$, we obtain:

$$\Delta W^* = DK^T(KK^T + \lambda C)^{-1} = DK^T A.$$

where $A \triangleq (\lambda C + \sum_{i=1}^M k_i k_i^T)^{-1}$ is the Precision Matrix. The update can be written as a sum of rank-1 updates:

$$\Delta W^* = \sum_{t=1}^M \delta_t k_t^T A.$$

### C.3. Error Decomposition

We analyze the effect of this global update on a specific target key $k_j$. The realized perturbation is:

$$\hat{\delta}_j = \Delta W^* k_j = \left(\sum_{t=1}^M \delta_t k_t^T A\right) k_j = \sum_{t=1}^M \delta_t (k_t^T A k_j).$$

Decomposing into the signal term ($t = j$) and interference term ($t \neq j$):

$$\hat{\delta}_j = \underbrace{\delta_j (k_j^T A k_j)}_{\text{Signal}} + \underbrace{\sum_{t \neq j} \delta_t (k_t^T A k_j)}_{\text{Crosstalk } \mathcal{E}_{cross}^{(j)}}.$$

### C.4. Bounding the Crosstalk

By the Triangle Inequality, the norm of the crosstalk error is:

$$\|\mathcal{E}_{cross}^{(j)}\|_2 \leq \sum_{t \neq j} \|\delta_t\|_2 \cdot |k_t^T A k_j|.$$

Let $\alpha_{tj} \triangleq |k_t^T A k_j|$. Assuming the metric space induced by $A$ is well-conditioned, $\alpha_{tj}$ represents the generalized inner product. Minimizing the upper bound requires minimizing $\alpha_{tj}$, which is equivalent to minimizing the pairwise cosine similarity between distinct keys $k_t$ and $k_j$.

## D. Detailed Empirical Analysis of Segment Independence

To further substantiate the theoretical claims in subsection 5.1, we provide a detailed experimental comparison between our Bayes-Chunk method and the baseline Fixed-Window approach.

## D.1. Global Statistical Analysis (KDE Plot)

We evaluate the semantic distinctness of segments generated by different partitioning strategies. For a given long-form sample $S$ partitioned into $N$ segments $\{s_1, s_2, \ldots, s_N\}$, we calculate the Intra-sample Mean Similarity $\bar{\sigma}(S)$ as:

$$\bar{\sigma}(S) = \frac{2}{N(N-1)} \sum_{1 \leq i < j \leq N} \text{CosSim}(\phi(s_i), \phi(s_j)).$$

where $\phi(\cdot)$ is the embedding function of the `Qwen3-Embedding-0.6B` model. We perform this calculation for all samples in the *EditEverything* dataset. The resulting Kernel Density Estimation (KDE) plot illustrates the probability density of $\bar{\sigma}$ across the dataset.

- **Bayes-Chunk:** Mean $\bar{\sigma} = 0.509$. The distribution is concentrated in the lower-similarity region, indicating that the segments are more semantically independent.

- **Fixed-Window:** Mean $\bar{\sigma} = 0.594$. The higher mean and right-shifted distribution suggest that fixed-size segments often contain redundant or highly correlated information, which leads to hardware/memory inefficiency and potential editing interference.

## D.2. Internal Key Orthogonality (Heatmap Analysis)

To observe the direct impact on the editing solver, we extract the actual anchor keys $k_t$ from the target editing layer of the `Llama-3.1-8B-Instruct` model.

**Extraction Protocol:** For a sequence of segments $\{s_1, \ldots, s_N\}$, we feed the cumulative context (Question + $\sum_{i=1}^{t} s_i$) into the model. The anchor key $k_t$ is defined as the hidden state at the last token of segment $s_t$ at the target layer $l$, specifically the tensor immediately preceding the output weight matrix $W_{out}$.

**Results:** We compute the pairwise cosine similarity matrix $M \in \mathbb{R}^{N \times N}$, where $M_{ij} = \frac{k_i^T k_j}{\|k_i\|_2 \|k_j\|_2}$.

- **Fixed-Window Heatmap:** Shows strong off-diagonal activation. This confirms that adjacent fixed-length chunks often produce highly similar keys, leading to the *overwriting* effect described in Theorem 5.1.

- **Bayes-Chunk Heatmap:** Displays a sharp diagonal structure with significantly suppressed off-diagonal values. By splitting the text at points of high surprisal, the hidden states are forced to undergo a state transition, resulting in keys that are more orthogonal in the latent space.

This visualization provides direct evidence that Bayes-Chunk satisfies the prerequisite for the crosstalk bound $\mathcal{E}_{cross} \to 0$, enabling stable mass-editing in long-form contexts.

# E. Proof of Causal Locality (Theorem 5.2)

In this section, we provide the rigorous derivation for the superior controllability of the immediate predecessor state.

## E.1. Problem Formulation

Let the gradient flow from the target loss $\mathcal{L}$ to an arbitrary hidden state $h_i^{(\ell)}$ be decomposed via the chain rule:

$$\nabla_{h_i^{(\ell)}} \mathcal{L} = \underbrace{\left( \frac{\partial \mathcal{L}}{\partial h_{t-1}^{(L)}} \right)^T}_{G_{\text{final}}} \cdot \underbrace{\frac{\partial h_{t-1}^{(L)}}{\partial h_i^{(\ell)}}}_{J_{i \to t-1}}.$$

where $G_{\text{final}}$ represents the gradient at the final layer readout, and $J_{i \to t-1}$ is the Jacobian of the intermediate layers.

*Proof.* We analyze the magnitude of the terms for $i = t - 1$ versus $i = t - k$.

**Step 1: The Magnitude of $G_{\textbf{final}}$.** Consider the cross-entropy loss $\mathcal{L} = -\log(\text{Softmax}(W_u h_{t-1}^{(L)}))_{y_t}$. The gradient w.r.t logits is $p - \mathbf{e}_{y_t}$. For a high-surprisal target (where the model initially predicts low probability $p_{y_t} \to 0$), the error vector norm approaches saturation:

$$\lim_{p_{y_t} \to 0} \|p - \mathbf{e}_{y_t}\|_2 \approx 1.$$

Assuming the unembedding matrix $W_u$ is full-rank with smallest singular value $\sigma_{\min} > 0$, we have a lower bound:

$$\|G_{\text{final}}\|_2 \geq \sigma_{\min}(W_u) \cdot \|p - \mathbf{e}_{y_t}\|_2 \geq \xi > 0.$$

**Step 2: Vertical Propagation (The Highway).** For the immediate position $i = t - 1$, the signal backpropagates vertically. In modern Transformers, residual branches are initialized with scaled variance ($W \sim \mathcal{N}(0, \frac{1}{2dL})$). Thus, the Lipschitz constant of the sub-layer function $F_m$ is small ($\lambda \ll 1$). The Jacobian through the residual stream $I + \frac{\partial F}{\partial h}$ has singular values clustered around 1:

$$J_{t-1 \to t-1} \approx I \implies \|G_{\text{final}} J_{t-1 \to t-1}\|_2 \approx \|G_{\text{final}}\|_2.$$

**Step 3: Horizontal Propagation (The Bottleneck).** For a distant position $i = t - k$, the signal must pass through the Attention mechanism: $h_{t-1} = \sum_j A_{t-1,j} V_j$. The Jacobian includes the attention weight term:

$$J_{t-k \to t-1} \approx A_{t-1, t-k} W_V.$$

Since $\sum_j A_{t-1,j} = 1$, for any specific distant token $k$, the attention weight is typically small ($A_{t-1, t-k} \ll 1$), acting as a damping factor $\gamma < 1$. Thus:

$$\kappa(t - k \to t) \approx \gamma \|G_{\text{final}}\|_2 \ll \|G_{\text{final}}\|_2.$$

**Conclusion.** Comparing the two terms:

$$\Delta \kappa_k \approx (1 - \gamma) \|G_{\text{final}}\|_2.$$

Since $\|G_{\text{final}}\|_2$ is bounded away from zero (Step 1) and $\gamma < 1$ (Step 3), it follows that $\Delta \kappa_k > 0$. This confirms that the residual stream provides a high-bandwidth control channel, while attention acts as an information filter. $\square$

## F. Sensitivity Analysis of Causal Controllability

In subsection 5.2, we posited that the immediate predecessor state provides the most efficient control over target predictions. Here, we provide the detailed experimental setup and extended results for the gradient sensitivity analysis.

### F.1. Experimental Setup

We randomly sample long-form instances from the *EditEverything* dataset and feed them into the `Llama-3.1-8B-Instruct` model. Let $h_{L,T}$ denote the hidden state of the $T$-th (last) token at the $L$-th (final) layer. To measure the intrinsic controllability of each position $(l, i)$, we define the loss as the norm of this final state: $\mathcal{L} = \|h_{L,T}\|_2$. We then compute the gradient sensitivity $G_{l,i}$ for every layer $l \in [1, L]$ and every token position $i$:

$$G_{l,i} = \left| \frac{\partial \mathcal{L}}{\partial h_{l,i}} \right|_2.$$

For cross-sample comparison, the sensitivities are normalized within each sample.

### F.2. Vertical vs. Horizontal Decay Analysis

The analysis in Figure 4(a) decomposes the gradient behavior into two orthogonal dimensions:

1. **Vertical Stability (Layer-wise):** The left sub-plot shows the mean sensitivity across layers for the target token. The curve is relatively flat, indicating that the gradient signal propagates through the residual stream with minimal attenuation. This supports our assumption that vertical propagation acts as a quasi-isometry.

2. **Horizontal Decay (Token-wise):** The right sub-plot illustrates the sensitivity as a function of the relative distance from the target token ($T - i$). A steep decline is observed: once the distance exceeds a few tokens, the sensitivity drops by orders of magnitude. This confirms that the attention mechanism acts as a bottleneck for *horizontal* control signals.

### F.3. Layer-Token Heatmap Visualization

To provide a holistic view, we visualize the full sensitivity matrix $M \in \mathbb{R}^{L \times T}$ as a heatmap (Figure 4(b)).

- **Observation:** The heatmap exhibits a prominent vertical band at the positions closest to the target token (the rightmost or *latest* columns in the sequence). This band maintains high intensity across almost all layers, while the rest of the map remains sparsely activated.

- **Implication:** This *vivid column* provides visual evidence for our Theorem 5.2: the most effective site for knowledge injection is not layer-specific, but rather position-specific. By targeting the immediate predecessors of the information we wish to update, we can minimize the update norm $\|\delta\|$ required to achieve the desired output, maximizing parameter efficiency and reducing collateral damage to distant context.

## G. Dataset Details

To validate the generalizability and robustness of AnyEdit++, we selected datasets that represent different levels of editing difficulty.

**EditEverything**: Reliably editing long texts requires maintaining logical chains and syntactic correctness, which standard triplet-based datasets cannot evaluate. We utilize the **EditEverything** benchmark, which aggregates high-complexity samples from seven diverse domains: *Mathematics, Code, Physics, Chemistry, Biology, News, and Poetry*. Unlike single-sentence edits, these tasks require the model to handle:

- *Structural Formatting*: Preserving code indentation or poetic stanzas after editing.

- *Logical Coherence*: Ensuring multi-step mathematical derivations remain valid after modifying an initial condition.

This serves as the primary testbed for our structure-aware segmentation, highlighting the failure cases of fixed-window approaches.

**CounterFact**: We use **CounterFact** (Meng et al., 2022a) to evaluate standard knowledge editing performance. It consists of counterfactual updates (e.g., changing *The capital of France is Paris* to *Rome*). This dataset allows us to measure:

- *Efficacy*: Success rate of the edit.

- *Locality*: Ensuring the edit does not affect unrelated knowledge.

Inclusion of CounterFact ensures that AnyEdit++ retains strong performance on fundamental editing tasks compared to existing baselines.

**UnKE**: We use **UnKE** (Deng et al., 2024) bridges the gap between rigid triplets and free-form text. It evaluates the model's ability to answer questions based on edited knowledge presented in unstructured formats. We use this to verify that our method can generalize edits beyond exact string matching, effectively updating the underlying semantic knowledge.

## H. Detailed Experimental Setup

### H.1. Implementation Details

Our implementation is built upon the official codebases of MEMIT and ROME. All experiments are conducted using the PyTorch framework and the HuggingFace Transformers library. The core editing algorithm follows the mass-editing flow, where the update is distributed across multiple layers.

### H.2. Hyperparameters for Target Vector Optimization

The computation of the target vector $z$ (represented as 'compute_z_bayes.py' in our codebase) is a critical step in our method. Unlike the sliding window approach used in baselines, we employ a surprisal-based adaptive segmentation strategy.

**Optimization Objective.** For each segmented semantic unit, we optimize the delta vector $\delta$ to minimize the negative log-likelihood (NLL) of the target sequence while constraining the norm of the change. The loss function $\mathcal{L}$ is defined as:

$$\mathcal{L} = \mathcal{L}_{NLL} + \lambda \frac{\|\delta\|_2}{\|z_{init}\|_2^2}.$$

where $z_{init}$ is the initial representation of the subject, and $\lambda$ is the weight decay factor. We use the **Adam** optimizer for this process.

**Adaptive Segmentation Configuration.** Our adaptive segmentation relies on the surprisal values (information content) of the target sequence. The specific hyperparameters used to determine boundaries are:

- **Boundary Density Ratio:** We set the number of segments $k$ proportional to the sequence length $L$. Specifically, $k = \lceil L/40 \rceil$. This ratio was empirically chosen to balance semantic coherence and computational efficiency.

- **Early Window Strategy:** To prevent generation collapse at the beginning of long sequences, we enforce a heuristic constraint: if no boundary is detected within the first $N = 3$ tokens based on the global top-$k$ surprisal sorting, we forcibly insert a boundary at the position with the maximum surprisal within this local window ($[0, N)$), replacing the boundary with the lowest surprisal score in the set.

### H.3. Global Hyperparameters

Table 6 lists the detailed hyperparameters used in our experiments.

*Table 6.* Hyperparameters used for MEMIT-ARE experiments.

| Parameter | Value |
|---|---|
| *Target Vector Optimization ($z$)* | |
| Optimizer | Adam |
| Learning Rate ('v_lr') | $5 \times 10^{-1}$ |
| Max Gradient Steps ('v_num_grad_steps') | 20 |
| Loss Threshold (Early Stopping) | $1 \times 10^{-2}$ |
| Weight Decay ($\lambda$) | $5 \times 10^{-1}$ |
| Norm Constraint Factor | $3.0 \times \|z_{init}\|$ |
| *Covariance Statistics* | |
| Covariance Dataset | Wikipedia (wikitext) |
| Sample Size ('mom2_n_samples') | 100,000 |
| Update Weight ('mom2_update_weight') | 15000 |
| *Segmentation Logic* | |
| Segments Count Formula | $\lceil \text{Length}/40 \rceil$ |
| Force Early Window Size ($N$) | 3 tokens |

### H.4. Covariance Statistics

Following the standard MEMIT setting, we estimate the key-value covariance matrix $C$ using a subset of the Wikipedia dataset. We collect second-moment statistics ('mom2') over 100,000 samples to ensure the edit minimizes interference with the model's general knowledge. The update is applied by solving the least-squares problem:

$$W_{new} = W_{old} + \Lambda(C + KK^T)^{-1}K^T.$$

where $K$ represents the stacked keys from the editing targets and $\Lambda$ is the residual distributed across layers.

```python
def compute_surprisal(model, tokenizer, question, answer):
    """
    Computes Bayesian Surprisal for each token in the answer
    conditioned on the question and preceding context.
    """
    model.eval()

    # 1. Construct full context
    full_text = f"{question} {answer}"
    enc = tokenizer(full_text, return_tensors="pt", return_offsets_mapping=True)
    input_ids = enc["input_ids"].to(model.device)

    # 2. Identify indices corresponding to the answer part
    # (Implementation detail: uses offset_mapping to align chars to tokens)
    answer_indices = find_answer_token_indices(enc, answer)

    # 3. Forward pass to get log-probabilities
    with torch.no_grad():
        logits = model(input_ids).logits
        # log_softmax over the vocabulary dimension
        log_probs = torch.log_softmax(logits, dim=-1)

    surprisals = []
    # 4. Calculate Surprisal: -log2 P(x_t | x_{<t})
    for pos_i in answer_indices:
        # Skip the first token of the sequence (no history)
        if pos_i == 0: continue

        # The target token at position `pos_i` is predicted
        # by the hidden state at `pos_i - 1`
        target_token_id = input_ids[0, pos_i]
        log_prob = log_probs[0, pos_i - 1, target_token_id]

        # Convert natural log to base-2 surprisal
        surprisal = -log_prob.item() / np.log(2.0)
        surprisals.append(surprisal)

    return np.array(surprisals)
```

*Listing 1.* Core implementation for computing token-level surprisal. We calculate the conditional probability of each token in the answer given the prefix (question and preceding answer tokens).

## I. Build Our QwQ-Edit Dataset

Our QwQ dataset was sampled from the `fql/qwq_long_cot_math_gsm_v1` dataset, which obtained original questions from the GSM-8K and MATH datasets and then constructed CoT responses using QwQ-32B. It comprises a total of 8.2k training examples and 2.04k test examples. We randomly selected 300 samples from the test set to form our QwQ-Edit dataset. An example of the QwQ-Edit dataset is shown in Figure A.

### I.1. Dataset Metric Definitions

To analyze the complexity and logical depth of the *QwQ-Edit* dataset compared to standard baselines, we defined specific metrics for text length and logical density.

**Logic Density Calculation**: We quantify the logical density of a solution by measuring the frequency of explicit reasoning markers. The *Logic Density* metric is defined as the number of reasoning keywords per 100 words in the generated output:

$$\text{Logic Density} = \frac{N_{\text{keywords}}}{N_{\text{words}}} \times 100.$$

where $N_{\text{words}}$ is the total word count of the solution text, and $N_{\text{keywords}}$ is the count of specific logical connectives identified

via case-insensitive pattern matching.

**List of Reasoning Keywords**: The calculation considers a set of 15 causal and procedural keywords commonly associated with chain-of-thought reasoning. The complete list of keywords used in our analysis is:

*therefore, because, thus, hence, consequently, implies, step, first, second, finally, so, then, since, result, leads to*

**Length Metrics**: Text length statistics are derived based on word-level tokenization (splitting by whitespace). *Total Length* refers to the sum of the word counts for both the problem statement (input) and the solution (output).

## J. Additional Data

Here we present full-text samples from QwQ-Edit mentioned above, providing a more intuitive understanding of the data.

---

**Sample 1 (Source: GSM8K) - Multi-step Arithmetic**

**Problem:**
A dog runs through a field and is joined by a group of 4 cats. 2 rabbits join each cat and 3 hares join each rabbit. How many animals are running through the field?

**Solution (Chain-of-Thought):**
Let's tackle this problem step by step. We start with a dog running through a field. So, that's one animal. Next, a group of 4 cats joins the dog. ... (omitted basic steps) ...
So, the number of hares is: 8 rabbits $* 3$ hares/rabbit $= 24$ hares. Adding these 24 hares to the current total of 13 animals, we get: $13 + 24 = 37$ animals.

*[Self-Correction/Review: Wait a minute, let me double-check that I haven't missed anything.]* We started with 1 dog, added 4 cats, then 2 rabbits per cat, and finally 3 hares per rabbit. That seems correct. Alternatively, perhaps there's a different way to approach this problem...

Total: $1 + 4 + 8 + 24 = 37$. I think this is solid.

**Final Answer**

$$\boxed{37}$$

---

*(Analysis: This sample features a lengthy verification process where the model re-calculates using a "tree structure" approach to confirm the initial result.)*

---

**Sample 2 (Source: MATH) - Combinatorics**

**Problem:**
In how many ways can a president and a 2-person committee be chosen from a group of 8 people (where the order in which we choose the 2 people doesn't matter)? (The president cannot be on the committee.)

**Solution (Chain-of-Thought):**
So I've got this problem here... Let's break this down. First, I need to choose a president... implies 8 ways. Next, I need to choose a 2-person committee from the remaining 7 people... $C(7, 2) = \frac{7!}{2!(7-2)!} = 21$.

Total ways $= 8 \times 21 = 168$.

*[Self-Correction/Review: Wait a minute, does that make sense?]* Another way to think about it is to first select the 2-person committee from all 8 people ($C(8, 2) = 28$), and then choose the president from the remaining 6 people (6 ways). $28 \times 6 = 168$.
*[Self-Correction/Review: Wait, that's the same result as before.]*
**Final Answer**

$$\boxed{168}$$

---

*(Analysis: The model explicitly tests two different combinatorial paths (President first vs. Committee first) to ensure consistency.)*

---

**Sample 3 (Source: GSM8K) - Algebra Generation**

**Problem:**
Francine has five full boxes of crayons and 5 loose crayons, and her friend has 27 loose crayons. They need to put all of their loose crayons in a box. How many more boxes do they need if Francine has a total of 85 crayons?

**Solution (Chain-of-Thought):**
...First, I need to figure out how many crayons are in each full box.

$$5 \times (\text{number of crayons per box}) + 5 = 85$$

Solving for crayons per box gives 16.
Total loose crayons: $5 + 27 = 32$. Dividing by box capacity: $32/16 = 2$.

*[Self-Correction/Review: Wait a minute, let me double-check that.]* Francine already has five full boxes... the question is asking how many MORE boxes... Yes, it seems correct.
**Final Answer**

$$\boxed{2}$$

**Sample 4 (Source: GSM8K) - Temporal Arithmetic**

**Problem:**
There were 80 people at a football game on Saturday. On Monday, 20 fewer people were at the football game... (Sequence of updates)... If their expected total audience... is 350, how many more people attended...?

**Solution (Chain-of-Thought):**
Let's list out the number for each day:

- Saturday: 80

- Monday: $80 - 20 = 60$

- Wednesday: $60 + 50 = 110$

- Friday: $80 + 60 = 140$

Total attendance = $80 + 60 + 110 + 140 = 390$. Difference = $390 - 350 = 40$.
**Final Answer**

$$\boxed{40}$$

**Sample 5 (Source: GSM8K) - Logic Trap**

**Problem:**
Heather is going to sew 150 aprons... She already sewed 13, today sewed 3x as many... How many should she sew tomorrow if she wants to sew half of the remaining number?

**Solution (Chain-of-Thought):**
...Today she sewed: $3 \times 13 = 39$. Total sewn so far: $13 + 39 = 52$. Remaining needed: $150 - 52 = 98$.
Target for tomorrow is half of remaining: $98/2 = 49$.

*[Self-Correction/Review: But wait, let's double-check that.]* After sewing 49 tomorrow, total sewn is $52 + 49 = 101$. Remaining is 49. So sewing 49 is exactly half of the 98 derived needed. Makes sense.
**Final Answer**

$$\boxed{49}$$

