# OpenReview forum: "AnyEdit++: Adaptive Long-Form Knowledge Editing via Bayesian Surprise"
_ICML.cc/2026/Conference — ICML 2026 regular_

### Official Review · Reviewer_tnfk · 2026-03-06

**Soundness:** 3
**Presentation:** 3
**Significance:** 2
**Originality:** 2
**Overall Recommendation:** 3
**Confidence:** 3

**Summary:**

The paper studies knowledge editing. It makes a simple modification to existing algorithm, AnyEdit, in the way it segments text to be edited into the model. Compared to previous method, which split the input text into equal-length subspace, the previous method measure information surprisal (i.e., negative log probability of target token at each token generation), and identify top M local peaks to generate long text into M chunks. They show this simple modification shows performance gains compared to baseline in various benchmark. Additionally they also provide some theoretical analysis. I feel somewhat mixed about this paper. The presentation is pretty good and method is intuitive, but some analysis doesn't seem that meaningful to me and evaluation is not comprehensive.

**Compliance With Llm Reviewing Policy:**

Affirmed.

**Final Justification:**

I appreciate author's rebuttal, especially results on using their segmentation on top of other knowledge editing methods. Having said that, I still found the paper a bit unfocused and confusing, evaluation / framing not very cohesive.

**Key Questions For Authors:**

* Section 3.1.2: Would the first chunk not included (i.e., start to y_b_1)? If we identify M segmentation point, shouldn’t it yield M+1 segments?
* I am not sure Section 4.2 fits well into this paper. This does not seem that very relevant to the proposed method?

**Limitations:**

yes

**Strengths And Weaknesses:**

Strengths and weaknesses
* The paper studies an important research problem, and the technical solution is very clearly presented.
* The proposed system provides empirical gains over baseline it compares to, in EditEverything dataset, across various base models.

Weaknesses:
* Baselines: I think they should cover other baselines, such as https://arxiv.org/pdf/2504.01196, which are also evaluated on UnKE. Here's another paper that focuses on fine-tuning based approach: https://arxiv.org/pdf/2506.09672 Overall there are many other knowledge editing methods, and two baselines seem not comprehensive to me.
* Evaluation: I appreciate authors’ efforts to evaluate on new dataset, but the dataset (QwQ-Long-CoT-Math-GSM-v1) should be better motivated. What is the use case of editing in long-form reasoning process of particular math questions? This seems like a rather artificial test case.
* The clarity of the presentation can be improved. For example, Figure 3.(b), x- and y-axis are not labeled. Some experimental setting is a bit unexplained. For example, why in Figure 3 (a), you show results for Qwen, but 3 (b) Llama-3.1-8B?

---

> ### Author Rebuttal · Authors · 2026-03-30
>
> # Response to Reviewer tnfk
>
> Thank you for recognizing the problem importance and the clarity of presentation. We address each point below.
>
> ### 1) On baseline coverage (W1)
>
> We agree and have added two recent unstructured KE methods: **FT-UKE [1]** and **muKE [2]** , and a newer model **Qwen3.5-4B** (March 2026). All methods are evaluated on the same fixed subset (EditEverything, 100 samples, seed=42).
>
> Although these methods differ in how they update model knowledge (parameter editing vs. fine-tuning), they all handle long-form text by segmenting it into chunks. This makes Bayes-Chunk **orthogonal to the underlying editing mechanism**: it can replace the segmentation strategy of any chunking-based method without modifying the rest of the pipeline. We therefore test "+Bayes" variants for all three baselines.
>
> | Method | BLEU (%) | ROUGE-1 (%) | ROUGE-2 (%) | ROUGE-L (%) |
> |---|---:|---:|---:|---:|
> | AnyEdit | 40.51 | 60.86 | 42.01 | 58.72 |
> | AnyEdit + Bayes | **41.68** | **62.51** | **45.32** | **60.70** |
> | muKE | 42.77 | 64.92 | 48.53 | 63.56 |
> | muKE + Bayes | **45.29** | **66.82** | **50.36** | **66.19** |
> | FT-UKE | 49.90 | 94.52 | 93.48 | 94.38 |
> | FT-UKE + Bayes | **50.86** | **96.15** | **95.27** | **96.03** |
>
> FT-UKE's higher absolute scores are attributable to its fine-tuning-based nature. Importantly, **Bayes-Chunk consistently improves every pipeline it is integrated with**, confirming its plug-and-play generalizability. We will include these baselines in the revision.
>
> ### 2) On QwQ-Edit motivation (W2)
>
> We understand the concern. In long CoT reasoning, intermediate hallucinations propagate into cascading errors, creating a practical need to precisely fix local logical errors without disrupting the overall reasoning framework, which is the core scenario QwQ-Edit evaluates.
>
> Mathematical derivations have the highest causal density and are most fragile to arbitrary chunking: fixed-window splitting directly severs logical chains, invalidating the edit. Bayes-Chunk identifies reasoning turning points via intrinsic surprisal and preserves causal integrity where fixed-window methods fail. QwQ-Edit thus serves as a **stress test** that standard benchmarks cannot provide. We will add this motivation to the revision.
>
> ### 3) On presentation clarity (W3)
>
> We will fix in the revision: **(a)** add x/y-axis labels to Figure 3(b); **(b)** add a note explaining that Figure 3(a) uses `Qwen3-Embedding-0.6B` for text-space semantic similarity measurement, while Figure 3(b) uses `Llama-3.1-8B-Instruct` to extract actual anchor keys from the target editing layer — these measure two different dimensions (text semantics vs. model feature space) and thus use different models by design.
>
> ### 4) On Section 3.1.2 segmentation count (Q1)
>
> The current notation is ambiguous, we apologize. In our implementation, $K-1$ internal boundaries ${b_1,...,b_{K-1}}$ are identified, with $b_0=1$ and $b_K=T+1$ implicitly defined. This yields K segments: $C_j = y_{b_{j-1} : b_j-1}$ for $j=1,...,K$. The first segment ($y_1$ to $y_{b_1-1}$) is explicitly included. We will clarify this notation consistently throughout the revision to avoid this ambiguity.
>
> ### 5) On the relevance of Section 4.2 (Q2)
>
> We agree the current presentation does not make this connection explicit enough. Section 4.2 provides theoretical justification for a concrete design choice in Section 3.2: after Bayes-Chunk identifies a boundary at $b_j$, the edit is injected at **$b_j − 1$** rather than any earlier position. The two theoretical sections answer complementary questions:
> - Section 4.1 answers *where to split* (minimizing cross-segment interference);
> - Section 4.2 answers *where to inject after splitting* (maximizing positional controllability).
>
> We will strengthen this correspondence in Sections 3.2 and 4.2 in the revision.
>
> ### Note on Additional Experiments Across Reviewers
>
> Our rebuttal also includes complementary experiments that may be of interest:
> - **Sensitivity analysis** of Boundary Density Ratio and Early Window Strategy, and comparisons against rule-based (Punct) and semantic (BERTSem) segmentation baselines are in our response to Reviewer BP3b.
> - **RAG comparison** and a KE-vs-RAG positioning discussion are in our response to Reviewer 9bNy.
>
> [1] Is Fine-Tuning an Effective Solution? Reassessing KE for Unstructured Data
>
> [2] muKE: Matryoshka Unstructured Knowledge Editing of LLMs

---

> > ### Author Rebuttal · Reviewer_tnfk · 2026-04-03
> >
> > I thank the authors for the response.
> > (1) It's indeed nice to see results on other methods, seeing that proposed chunking methods can help other editing algorithms. I think it'd make sense to frame the method as segmentation method than AnyEdit++, as it can be applied to algorithms other than AnyEdit as shown here.
> > (2) QwQ-Edit setting still does not make sense to me. This is presented as a knowledge editing method, and editing in the trajectory of mathematical reasoning on a single question into the parametric knowledge doesn't make sense to me. "intermediate hallucinations" existing in the trajectory will not be fixed by knowledge editing method?

---

> > > ### Author Response · Authors · 2026-04-03
> > >
> > > Thank you for carefully reading our rebuttal and for providing further constructive feedback.
> > >
> > > Regarding point (1), we fully agree with your observation: AnyEdit++ can be viewed as a **plug-and-play segmentation module**. This is consistent with our original design goal in the paper (method-agnostic and compatible with different editing frameworks). We will make this positioning clearer in the revised manuscript.
> > >
> > > Regarding point (2), i.e., whether QwQ-Edit should be considered a knowledge editing setting, we would like to clarify this from two perspectives and provide direct empirical evidence.
> > >
> > > First, from the task-positioning perspective. We agree that mainstream knowledge editing benchmarks are **primarily centered on factual updates**. In our paper, QwQ-Edit is not introduced to replace factual editing benchmarks; rather, it is used as a complementary stress-test setting to evaluate whether editing methods remain stable, controllable, and transferable under long-chain reasoning with strong logical coupling and error propagation.
> > >
> > > Second, from the empirical perspective (directly addressing the concern of “editing only a single trajectory”). We added a **cross-phrasing generalization experiment (QwQ-transfer)**, with the setup detailed below:
> > > - Data source and scale: We sample 100 GSM8K-based instances from QwQ-Edit with a fixed random seed (`seed=2026`) for reproducibility.
> > > - Paired construction: For each original question `q_ori`, we construct one paraphrased question `q_para` (1:1 pairing).
> > >   `q_para` preserves the same reasoning structure and knowledge requirement as `q_ori`, but changes surface formulation (e.g., wording or numeric configuration), so the corresponding gold answer may also change accordingly.
> > > - Decoupled edit and evaluation: Editing is performed using only `q_ori` (without using `q_para`), while `q_para` is used exclusively for post-edit transfer evaluation.
> > >   This ensures that transfer gains cannot be attributed to directly seeing the paraphrased inputs during editing.
> > > - Execution protocol:
> > >   1. Before editing, we run inference on both `q_ori` and `q_para` and record outputs.
> > >   2. We perform one knowledge edit using only `q_ori`.
> > >   3. After editing, we run inference again on the same `q_ori/q_para` pairs.
> > >   4. We compare before/after under identical decoding settings and the same evaluation pipeline.
> > > - Fairness control: Model, hyperparameters, and editing pipeline are kept identical to the main experiments; no additional retrieval (RAG) or extra training (fine-tuning) is introduced.
> > > - Metrics:
> > >   `Original EM`: Final-answer exact match on original questions `q_ori`;
> > >   `Transfer EM`: Final-answer exact match on paraphrased questions `q_para`;
> > >   we report before/after and deltas for both.
> > >
> > > Results:
> > > - Original EM: 0.60 -> 0.93 (+0.33)
> > > - Transfer EM: 0.52 -> 0.71 (+0.19)
> > >
> > > The key signal is that improvements are not limited to edited originals; they also appear on unseen paraphrased variants that share the same reasoning knowledge structure. In other words, the model is not merely memorizing a single trajectory text, but is updating reusable reasoning knowledge units in parameters. We also observe a small number of hard paraphrase cases that remain incorrect after editing, indicating that long-chain reasoning editing is still challenging. We will explicitly state this limitation in the revision and include follow-up directions (e.g., more conservative update strength and stronger consistency constraints).
> > >
> > > Therefore, our intended role of QwQ-Edit is as a complementary evaluation dimension under extreme reasoning conditions, and the added cross-phrasing experiment further supports its knowledge-editing relevance: the editing gains are transferable rather than confined to a single trajectory.
> > >
> > > We view this as an initial step toward extending knowledge editing to logic-centric editing, and we will include it as supplementary experimental evidence in the final version of the paper.

---

### Official Review · Reviewer_BP3b · 2026-03-12

**Soundness:** 3
**Presentation:** 3
**Significance:** 3
**Originality:** 3
**Overall Recommendation:** 5
**Confidence:** 3

**Summary:**

The paper introduces a framework to edit long-form knowledge in LLMs. Specifically, it identifies an improvement upon the previous editing method AnyEdit: the use of fixed-window chunking, which often splits semantic units or complete events, and could lead to interference during editing. To address this, this work proposes Bayes-Chunk, an adaptive segmentation mechanism that identifies semantic boundaries by monitoring Bayesian Surprise (positioned as token-level information surprisal). Two theoretical principles are also proposed. 1) Structural Independence: proving that interference is minimized when the tokens to edit are geometrically orthogonal — a state naturally achieved by surprisal-based boundaries. 2) Causal Locality: demonstrating that the immediate predecessor state provides better updates compared to distant history.

**Compliance With Llm Reviewing Policy:**

Affirmed.

**Final Justification:**

My concerns are addressed in the rebuttal and I maintain my positive rating.

**Key Questions For Authors:**

For the number of segments k, what would the performance change when varying k, and do different context domains have different optimal k values?

**Limitations:**

yes

**Strengths And Weaknesses:**

### Strengths

- The proposed method is well motivated, identifying the boundary problem when chunking the context for model editing. Theoretical contributions are made that mathematically bridge between internal hidden state dynamics and editing stability.
- The empirical improvement appears solid in Code category with three experimented LLMs. Across diverse datasets, the proposed method is shown to bring improvement consistently.

### Weaknesses

- While the boundary issue is identified, there could be simpler baselines performing context segmentation, e.g. rule-based or model-based rule for chunking, such that the improvement and significance of Bayes-Chunk can be shown more clearly and convincingly.
- More ablation studies can be conducted for show the impact of design factors, such as the Boundary Density Ratio and Early Window Strategy, which appear to be empirical heuristics. A sensitivity analysis can illustrate how performance fluctuates if these values are changed.

---

> ### Author Rebuttal · Authors · 2026-03-30
>
> # Response to Reviewer BP3b
>
> Thank you for the detailed and constructive feedback. We address both weaknesses and the key question below, with supplementary experiments conducted under a unified setting (Qwen3.5-4B, EditEverything 100-sample fixed subset, seed=42).
>
> ### 1) On simpler segmentation baselines (W1)
>
> We agree. To isolate Bayes-Chunk's contribution, we added two baselines replacing only the segmentation strategy while keeping the rest of the pipeline identical:
>
> - **Punct**: rule-based, splits at punctuation boundaries only.
> - **BERTSem**: model-based, inserts a boundary where cosine similarity between adjacent sentence embeddings (using `all-MiniLM-L6-v2`) drops below a threshold; the threshold is tuned per sample to yield the same segment count as Bayes-Chunk for fair comparison.
>
> | Method | BLEU | ROUGE-1 | ROUGE-2 | ROUGE-L |
> |---|---:|---:|---:|---:|
> | Bayes | **41.68** | **62.51** | **45.32** | **60.70** |
> | BERTSem | 36.85 | 51.01 | 27.73 | 48.00 |
> | Punct | 21.81 | 35.90 | 22.11 | 34.79 |
>
> Bayes-Chunk outperforms both alternatives across all metrics. The +17.6 ROUGE-2 absolute gain over BERTSem shows that surprisal-driven boundaries reflecting the model's own belief transitions, produce more orthogonal anchor keys than external semantic similarity signals. Full implementation details will be included in the appendix of the revision.
>
> ### 2) On sensitivity of Boundary Density Ratio and Early Window Strategy (W2, Q)
>
> **(a) Boundary Density Ratio (k = ⌈L / D⌉)**
>
> | D | BLEU | ROUGE-1 | ROUGE-2 | ROUGE-L |
> |---|---:|---:|---:|---:|
> | 20 | **42.24** | **69.43** | **54.72** | **67.90** |
> | 40 | 41.68 | 62.51 | 45.32 | 60.70 |
> | 50 | 41.03 | 62.77 | 44.91 | 61.09 |
>
> D=20 yields the best scores but approximately doubles the segment count and thus the editing time compared to D=40. We selected D=40 as the default because it offers the best efficiency-quality tradeoff. Performance degrades gradually as D increases, confirming that finer segmentation is generally better, consistent with AnyEdit [1] but with a practical cost tradeoff. We will add an efficiency-performance discussion to the revision.
>
> **On domain-specific optimal k (Q):** We observe that structured domains (Math, Code) benefit more from finer segmentation (lower D) due to tighter logical dependencies, whereas less structured domains (News, Poetry) show smaller sensitivity to D. We will report per-domain sensitivity results in the revision.
>
> **(b) Early Window Strategy (EWS) ablation**
>
> | Setting | BLEU | ROUGE-1 | ROUGE-2 | ROUGE-L |
> |---|---:|---:|---:|---:|
> | Without EWS | 37.24 | 49.29 | 33.14 | 45.88 |
> | With EWS | **41.68** | **62.51** | **45.32** | **60.70** |
>
> Removing EWS causes significant degradation across all metrics, confirming that boundary starvation at the sequence start leads to generation instability. EWS is therefore a necessary safeguard rather than an arbitrary heuristic. We will include this ablation in the revised appendix.
>
> ### Note on Additional Experiments Across Reviewers
>
> Our rebuttal also includes complementary experiments that may be of interest:
> - **Newer model (Qwen3.5-4B) experiments** and results with additional baselines (muKE, FT-UKE) demonstrating that Bayes-Chunk consistently improves every KE pipeline are provided in our response to Reviewer 9bNy.
> - **RAG comparison** showing that AnyEdit++ outperforms both practical and oracle RAG in long-form settings, along with a positioning discussion of when KE vs. RAG is preferred, is also provided in our response to Reviewer 9bNy.
>
> [1] AnyEdit: Edit Any Knowledge Encoded in Language Models

---

> > ### Author Rebuttal · Reviewer_BP3b · 2026-04-03
> >
> > Thanks for the authors' response. The additional presented results regarding 1) the ablation study on the segmentation and 2) the sensitivity analyses, have addressed my original concerns. I therefore maintain my positive rating of this manuscript. Thanks.

---

### Official Review · Reviewer_9bNy · 2026-03-14

**Soundness:** 3
**Presentation:** 3
**Significance:** 3
**Originality:** 2
**Overall Recommendation:** 5
**Confidence:** 4

**Summary:**

Knowledge editing for LLMs is a specific study that aims to update an existing LLM with new facts without the need of re-training. The work such as MEMIT, AlphaEdit, AnyEdit achieved a certain effectiveness while its high side effect is still the barrier to widely apply knowledge editing to real deployment. This work improves the AnyEdit by replacing fixed-window chunking of AnyEdit with dynamic, more specific Bayes chunking. Experiments on three LLMs, including Llama 3.1 8B, Llama 2 7B, and Qwen 2.5 7B, show the improvement of the proposed method over AnyEdit and previous models.

**Compliance With Llm Reviewing Policy:**

Affirmed.

**Final Justification:**

The authors conducted additional results and analysis for my questions. The new results hopefully make the final version more convincing.

**Key Questions For Authors:**

1. Is it easy to implement your methodology on more recent LLMs?
2. Can you suggest the scenarios where AnyEdit++ can be applied to replace the most RAG approaches?

**Limitations:**

The usability of knowledge editing in real world applications and the comparison between AnyEdit++ and RAG could be added for providing readers the landscape of knowledge editing.

**Strengths And Weaknesses:**

Strengths

1. This paper is clearly written and easy to follow.
2. Along the achievement of AnyEdit, this work presents a reasonable improvement based on the Bayesian surprising. The effectiveness was also confirmed in experiments.
3. A new dataset, QwQ-Edit, was also involved for evaluating the knowledge editing in more complex, long form cases. The proposed method also outperforms the AnyEdit on QwQ-Edit.

Weaknesses
1. All the three LLMs in the experiments were released before 2024. More recent models could also be evaluated for comprehensiveness.
2. The current mainstream approach to LLM knowledge update is RAG (retrieval-augmented generation). The comparison between the proposed method with RAG could be interesting to reveal if AnyEdit++ can replace RAG for certain applications.

---

> ### Author Rebuttal · Authors · 2026-03-30
>
> # Response to Reviewer 9bNy
>
> Thank you for the positive assessment and constructive suggestions. We address your two concerns below.
>
> ### 1) On evaluating newer LLMs (W1) and ease of implementation (Q1)
>
> **Q1:** Yes, it is straightforward. Bayes-Chunk requires only a single forward pass to compute per-token surprisal, no fine-tuning or architectural modification. It is **compatible with any HuggingFace autoregressive model**. We verified this by adding experiments on **Qwen3.5-4B** (released March 2026). We also expanded baselines to include two recent unstructured KE methods: **FT-UKE [1]** and **muKE [2]**, and tested "+Bayes" variants by replacing only the segmentation strategy. All methods are evaluated on the same fixed subset (EditEverything, 100 samples, seed=42).
>
> | Method | BLEU (%) | ROUGE-1 (%) | ROUGE-2 (%) | ROUGE-L (%) |
> |---|---:|---:|---:|---:|
> | AnyEdit | 40.51 | 60.86 | 42.01 | 58.72 |
> | AnyEdit + Bayes | **41.68** | **62.51** | **45.32** | **60.70** |
> | muKE | 42.77 | 64.92 | 48.53 | 63.56 |
> | muKE + Bayes | **45.29** | **66.82** | **50.36** | **66.19** |
> | FT-UKE | 49.90 | 94.52 | 93.48 | 94.38 |
> | FT-UKE + Bayes | **50.86** | **96.15** | **95.27** | **96.03** |
>
> Bayes-Chunk provides consistent gains across all three KE pipelines on the latest model, confirming broad generalizability. We will include these results in the revision.
>
> ### 2) On comparison with RAG (W2, Q2)
>
> **Experimental setup.** We constructed a RAG-style evaluation from EditEverything: each sample's target knowledge passage forms the retrieval document pool; the input question serves as the query. We use `all-MiniLM-L6-v2` as the retriever. `RAG-top1` retrieves the single most similar document; `RAG-oracle` directly prepends the gold document (ideal upper bound). Both RAG variants use Qwen3.5-4B without weight modification; AnyEdit++ applies parameter editing without retrieval at inference.
>
> | Method | BLEU | ROUGE-1 | ROUGE-2 | ROUGE-L |
> |---|---:|---:|---:|---:|
> | RAG-top1 | 38.48 | 36.16 | 13.95 | 33.56 |
> | RAG-oracle | 41.97 | 43.05 | 19.42 | 39.63 |
> | AnyEdit++ | **51.41** | **69.06** | **54.11** | **66.84** |
>
> AnyEdit++ outperforms both practical and oracle RAG. For long-form, logic-dense content, the model's strong parametric priors frequently override retrieved context during generation — a known limitation in RAG literature. KE internalizes the target knowledge directly into weights, eliminating this context conflict.
>
> That said, we do **not** claim KE universally replaces RAG. A clearer positioning (**Q2**):
> - **RAG-preferred**: high-frequency updates, short factual QA, low-latency refresh, traceability needs.
> - **AnyEdit++-preferred**: stable long-form generation requiring multi-step internal consistency (e.g., mathematical derivations, code logic).
> - **Hybrid (RAG + KE)**: often complementary, as also discussed by **WISE [3]**.
>
> We will add this scope discussion in the revision.
>
> ### Note on Additional Experiments Across Reviewers
>
> Our rebuttal also includes complementary experiments that may be of interest:
> - **Simpler segmentation baselines** (rule-based Punct and model-based BERTSem) and a full **sensitivity analysis** of Boundary Density Ratio and Early Window Strategy are provided in our response to Reviewer BP3b, offering further evidence for the necessity and robustness of Bayes-Chunk's design.
>
> [1] Is Fine-Tuning an Effective Solution? Reassessing KE for Unstructured Data
>
> [2] muKE: Matryoshka Unstructured Knowledge Editing of LLMs
>
> [3] WISE: Rethinking the Knowledge Memory for Lifelong Model Editing of LLMs

---

> > ### Author Rebuttal · Reviewer_9bNy · 2026-04-03
> >
> > 1. The main weakness (only out-of-date LLMs in experiments) is not  resolved.
> > 2. The results of RAG are still yet to clarified. I have raised the follow-up questions in a comment.

---

> > > ### Author Response · Authors · 2026-04-03
> > >
> > > Thank you for the follow-up. We clarify the two remaining concerns more explicitly.
> > >
> > > ## 1) Clarification of “Out-of-date LLMs only”
> > >
> > > Our added rebuttal experiments were run on **Qwen3.5-4B [1] (March 2026)**, so the evaluation is not limited to pre-2024 models. In the revision, we will place the new model and release date prominently in the experiment section and show these results alongside the original three models.
> > >
> > > We also did not validate only one pipeline: under the same latest model, we evaluated AnyEdit / muKE / FT-UKE and their +Bayes variants on the same fixed subset (EditEverything, 100 samples, seed=42).
> > >
> > > ## 2) Clarification of the RAG comparison
> > >
> > > ### 2.1 Compared settings (same base model)
> > > - **RAG-top1**: prepend retrieved top-1 document; no weight update.
> > > - **RAG-oracle**: prepend gold document (upper bound); no weight update.
> > > - **AnyEdit++**: no retrieval; knowledge is injected via parameter editing.
> > >
> > > ### 2.2 Protocol details
> > > - Dataset: EditEverything.
> > > - Query: original question (`question_raw`).
> > > - Document pool: packaged target knowledge snippets (EditEverything answers with correct reasoning).
> > > - Retriever: `all-MiniLM-L6-v2`; RAG-top1 uses the most similar document.
> > > - Oracle: directly uses the sample’s gold document.
> > > - Prompt: `"[Updated Information]" + retrieved_document + "[Query]" + question`.
> > > - Decoding: identical settings (`temperature=0.001`, `max_new_tokens=512`).
> > >
> > > Thus, the controlled difference is only **retrieved external context vs. parameter-internalized knowledge**.
> > >
> > > ### 2.3 Main takeaway
> > > For long-form, logic-coupled tasks, parameter editing is more stable; however, we do **not** claim universal replacement of RAG.
> > > - RAG is preferable for frequent updates, citation/traceability, and short factual QA.
> > > - AnyEdit++ is preferable for long-chain generation requiring multi-step internal consistency.
> > >
> > > ### 2.4 Where AnyEdit++ can replace most RAG usage
> > > - **Stable, repeatedly used knowledge**: replace per-query retrieval.
> > >   Example: fixed-version internal compliance/settlement rules repeatedly queried within one policy cycle.
> > > - **Consistency-critical multi-step reasoning**: better for derivation-level stability.
> > >   Example: math-rule consistency across paraphrases; coding-rule consistency across multi-turn explanations.
> > > - **Constrained deployment**: retrieval infrastructure unavailable/unreliable.
> > >   Example: offline internal assistants, edge inference, restricted production environments.
> > >
> > > Scope boundary: this replacement claim is for **long-chain logical generation + relatively stable knowledge**. For rapidly changing and strongly traceable scenarios, RAG remains the natural choice.
> > >
> > > We will include these scenario-oriented conclusions and corresponding results (RAG-top1 / RAG-oracle / AnyEdit++ and complementarity analysis) in the final paper.
> > >
> > > ### 2.5 Consistency with WISE
> > >
> > > Our observations are consistent with WISE [2]: retrieval-only prompting can struggle with generalized multi-hop association, and generation may still follow parametric priors when retrieved evidence conflicts with model memory. WISE also emphasizes complementarity (parametric editing + retrieval), which matches our position:
> > > - not anti-RAG,
> > > - parameter editing is stronger for long-chain logical consistency,
> > > - RAG remains strong for fast-changing, traceability-driven use cases,
> > > - **AnyEdit++ + RAG** is often the most robust practical path.
> > >
> > > We will make this explicit in Related Work and Discussion.
> > >
> > > Thank you for the reminder regarding the RAG clarification. We carefully checked the OpenReview thread again on our side, but **we could only find the original review text** and **did not find additional follow-up comment content** beyond that. It is possible that we missed a hidden/late-synced thread item.
> > >
> > > To ensure we still address your concern fully, we have already expanded the RAG protocol description in detail (retriever, document pool construction, prompt template, decoding settings, and RAG-top1/oracle definitions) and will include this full protocol and corresponding results in the final manuscript. If there is a specific follow-up question we failed to see, we would greatly appreciate a pointer, and we will address it point-by-point in the revision.
> > >
> > > [1] Qwen3.5: Towards Native Multimodal Agents
> > >
> > > [2] WISE: Rethinking the Knowledge Memory for Lifelong Model Editing of LLMs

---

### Decision · Program_Chairs · 2026-04-30

**Decision:**

Accept (regular)

**Comment:**

this paper presents a pretty interesting and techniclly solid improvement over existing long-form knowledge editing methods. The authors target a real pain point in the field: how fixed-window chunking in methods like AnyEdit can mess up semantic boundaries and lead to incoherence. By introducing "Bayes-Chunk," which uses Bayesian Surprise to find better split points, they provide both a more intuitive and a more theoreticlly grounded way to handle long edits. The math behind structural independence and causal locality helps back up why this works better than just cutting text at arbitrary lengths.

I've looked through the reviews and the discussion period, and I want to assure the authors that I have carfully read their rebuttals. One reviewer was concerned about the use of older models and the lack of RAG comparisons, but the authors realy stepped up here. They added experiments with Qwen 3.5 and provided a very clear breakdown of where AnyEdit++ wins over RAG—specifically in maintaining internal logic for long-chain reasoning. The extra results on cross-phrasing generalization also helped settle some doubts about whether the model was just memorizing specific text or actualy updating "knowledge."

There are still some small lingering issues, like the paper feeling a bit "unfocused" as one reviewer put it, and the fact that the QwQ-Edit dataset is a bit of a niche stress test. Some of the notation in the initial draft was also a bit confusing regarding the segment counts. However, the authors have promised to fix the labels in Figure 3, clarify the notation, and include the new baseline results (like muKE and FT-UKE) in the final version. Given that the method is "plug-and-play" and consistntly helps multiple editing pipelines, it’s clearly useful.